# Self-assessed health status and obesity vulnerability in rural Louisiana: A cross-sectional analysis

Santosh Pathak[1‡], Hua Wang[1‡], Katherine Seals[2], Naveen C. Adusumilli[1]*, Denise Holston[2]

**1** Department of Agricultural Economics and Agribusiness, Louisiana State University (LSU) and LSU Agricultural Center, Baton Rouge, Louisiana, United States of America, **2** School of Nutrition and Food Sciences, Louisiana State University, Baton Rouge, Louisiana, United States of America

‡ SP and HW share first authorship on this work.
* nadusumilli@agcenter.lsu.edu

## Abstract

Rural communities are resource-constrained and at higher risk of obesity and obesity-related conditions. Thus, studying self-assessed health status and underlying vulnerabilities is critical to provide insights to the program planners for effective and efficient planning of obesity prevention programs. This study aims to investigate the correlates of self-assessed health status and subsequently determine the obesity vulnerability level of residents in rural communities. Randomly sampled data were obtained from in-person community surveys in three rural Louisiana counties–East Carroll, Saint Helena, and Tensas–in June 2021. The association of social-demographic factors, grocery store choice, and exercise frequency with self-assessed health was investigated using the ordered logit model. An obesity vulnerability index was constructed using the weights obtained from the principal component analysis. The results show that gender, race, education, possession of children, exercise frequency, and grocery store choice significantly influence self-assessed health status. Around 20% of respondents fall into the most-vulnerable segment and 65% of respondents are vulnerable to obesity. The obesity vulnerability index ranged from -4.036 to 4.565, indicating a wide heterogeneity in the vulnerability level of rural residents. The findings show that the self-assessed health status of rural residents is not promising along with a high level of vulnerability to obesity. The findings from this study could serve as a reference in the policy discussion regarding an effective and efficient suite of interventions in rural communities to address obesity and promote well-being.

## Introduction

Obesity is an emerging global epidemic and a public health concern because of its association with poor health outcomes [1–3]. Obesity is disproportionately prevalent in the US and rural communities suffer higher (~34.2%) and at an increasing pace compared to their urban counterparts (~28.7%) [4–6]. Since, a multitude of factors like genes, poor diet choices, limited

**Data Availability Statement:** An anonymized data set underlying the results described in the manuscript has been included as a Supporting Information file.

**Funding:** D.H. received funding from the Centers for Disease Control and Prevention (https://www.cdc.gov/) for this project under cooperative agreement number 58DP006570. The funders had no role in study design, data collection and analysis, decision to publish, or preparation of the manuscript.

**Competing interests:** The authors have declared that no competing interests exist.

exercise frequency, limited access to health care services, and social environment cause and influence obesity, addressing this epidemic is challenging [2, 7]. The aforementioned factors influencing obesity are more conducive in rural areas [3], thus further exacerbating obesity prevention efforts.

To narrow the urban-rural gap in obesity prevalence, it is necessary to investigate the relationship between obesity risk factors and self-assessed health (SAH) at the community level. Furthermore, understanding how social, demographic, and behavioral factors operate within a rural community is critical to developing effective interventions to address obesity. Utilizing data on established risk and protective factors, a vulnerability index could serve as an important tool for decision-makers [8] and health practitioners seeking to understand rural communities' vulnerability to obesity. Such studies facilitate the development of equitable solutions for public health practitioners working to prevent obesity in rural communities. Because rural communities are often constrained by resources, expertise, and infrastructure, it is important for local decision-makers to have access to practical data that describes the health status and inequities of a community to facilitate precise targeting of available resources. It is also important for decision-makers to understand how individuals evaluate their health in general and how associated risk and preventative factors operate and are contextualized within a community. It is because SAH status is an important health indicator and predictor of future morbidity and mortality [9].

The existing studies on SAH have centered their analyses on how and to what extent gender disparity [10], healthcare access [11], digital divide [12], financial strain [13], and fuel poverty [14] influence SAH. However, studies that focus on rural residents regarding the determinants of SAH are only a few. Furthermore, despite obesity prevalence rapidly increasing in rural areas, to our knowledge, there is not any study that explores the vulnerability level of households or individuals concerning obesity. The index-based measure to assess the vulnerability level is widely used and provides a picturesque of underlying exposure to risks and the level of preventive efforts of households or communities [8, 15]. Such index-based measure also provides objective criterion that aids the planning and implementation of obesity-related interventions. Thus, we attempt to fill this gap in the literature by investigating two major research questions: *i*) What factors influence SAH status in rural areas? *ii*) What is the vulnerability level of the rural residents in terms of obesity? To this end, we employ the ordinal logit regression model and the principal component analysis on data from community surveys in three rural counties in Louisiana, USA. The results show that grocery store choice and exercise frequency significantly influence SAH status, besides sociodemographic variables such as gender, education, and interaction between age and race. Furthermore, our findings show that around two-thirds of the respondents are vulnerable to obesity, thus necessitating prompt preventative actions to promote healthy well-being in rural Louisiana.

This study adds to the existing body of literature, including Au and Johnston [16], Dowd and Zajacova [17], Gray et al. [18], Holston et al. [19], Karnes et al. [20], Hill et al. [21] and Myers et al. [22] on SAH, obesity, and rural health, in three ways. First, we explore the influence of grocery store choice, exercise frequency, Supplemental Nutrition Assistance Program (SNAP) participation, and employment status on the SAH status in rural settings. The information about the influencing factors of SAH is important for health practitioners and local decision-makers to identify behavioral and social risk factors prevalent at the community level. Second, we introduce the obesity vulnerability index (OVI) that is relevant to the program planners and policymakers to make obesity prevention programs more targeted. The information about OVI and associated influencing factors could assist health practitioners, legislators, and local leaders to better target funding, precision interventions, and policies when addressing obesity in rural communities. Third, while health surveillance data is available at the state

and county level, rural county-level data often lack representative samples creating a unique challenge when decision-makers are tasked with addressing public health issues. Using representative data, we aim to assist public health researchers and practitioners by identifying behavioral and socio-environmental characteristics of rural communities that put members at risk for obesity.

## Data and methods

### Survey design and data collection

Population-level data is ideal to study overall health status, obesity prevalence, and underlying population vulnerabilities; however, such data can be expensive and difficult to obtain, especially in rural communities. Utilizing representative survey data is a viable option to assess overall SAH and associated risk factors. This research uses representative cross-sectional data from the Rural Eastern Louisiana Food Accessibility and Active Transportation (RELFA) survey to investigate predictors of SAH and determine obesity vulnerability in three rural Louisiana counties: East Carroll, Saint Helena, and Tensas (Fig 1). These three counties were selected because the obesity prevalence rate is over 40% and they remain priority areas of the Centers for Disease Control and Prevention (CDC) to address obesity. All survey development and data collection processes were guided by the principles of Community Based Participatory Research (CBPR) that prioritizes community engagement throughout all phases of research, including instrument development, data collection, and dissemination of results, to achieve equitable research [23]. The questionnaire for the REFLA survey was developed in consultation with experts in nutrition and transportation, extension agents, community members, and bike advocacy leaders. Before formally launching the survey, a pilot test of the questionnaire was conducted with the community members in the study area to ensure comprehensibility.

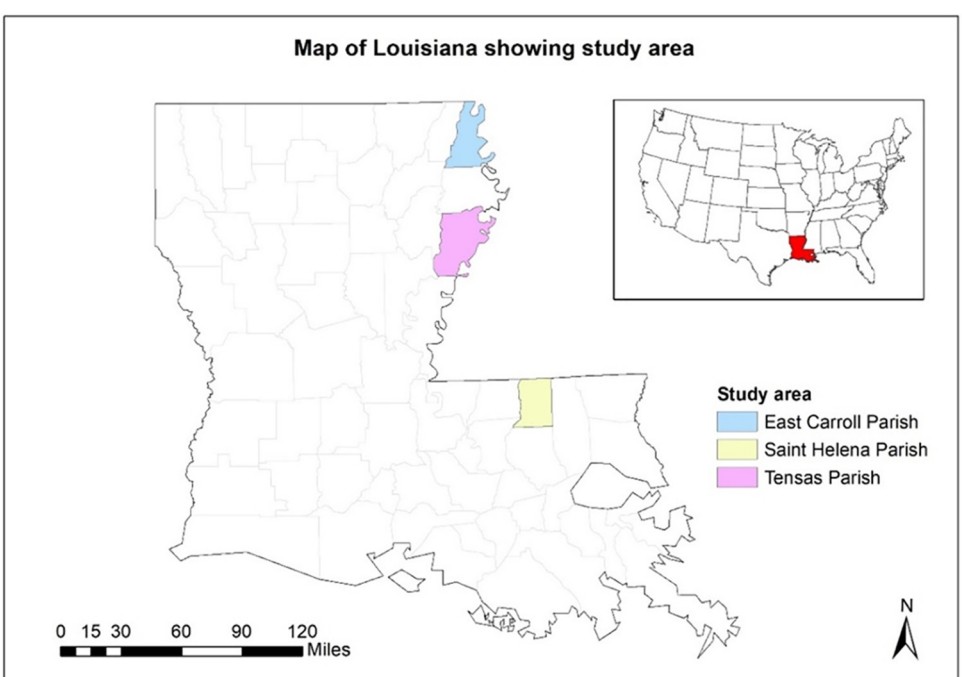

**Fig 1. Map of Louisiana showing three study parishes.** The inset map shows Louisiana in the contiguous United States.

We used population data from the 2019 American Community Survey [24] to determine the number of sample households in each community. To ensure survey responses were representative of the three rural counties under study, a probability sample ($n$ = 811) using an online random number generator (https://www.checkmarket.com/sample-size-calculator/) was drawn from a comprehensive list of residential United States Postal Service (USPS) addresses for each community. Because data were collected during the COVID-19 pandemic, data collection was initiated with postcards mailed to each randomly sampled address and included a website link and QR code to access the survey online. Following postcard data collection, in-person community data collection events were held in each community to survey all remaining addresses in the sample ($n$ = 779). Altogether 51 community members were recruited by research and extension staff to go door-to-door to collect surveys in their community over two consecutive weekends in June 2021. Data collectors were trained by the research team on data collection protocols and survey administration and were provided with data collection supplies, including iPads that could be used without the internet, paper surveys, and personal protection equipment (PPE).

To ensure randomization, a replacement scheme was employed if the data collector found that the originally sampled address was vacant, deemed dangerous, or a business. Participants were eligible to participate in the survey if they lived in the parish of interest, were at least 18 years of age, and did some of the food shopping in the household. Survey refusals were not included in the replacement scheme. Written informed consent was obtained before proceeding with the questionnaire, and surveys were either collected via iPad using the offline Qualtrics application or on paper. Survey participants received a tote bag with promotional items when they completed the survey. Further details about the survey instrument development and data collection protocols have been published in Seals et al. [25].

The RELFA survey elicited information about food shopping behaviors, perceptions of local food environments, active transportation behaviors, perceptions of those who walk and bike for transportation, the household ownership of working vehicles and bicycles, SAH, and general demographics to inform obesity prevention program strategies. The LSU Agricultural Center Institutional Review Board (IRB) approved the study procedures including the survey instrument (IRB# HE20-24) on May 15, 2020.

## Variables

SAH is a commonly used measure in clinical practice, research, policy, and general population surveys as it provides a valid and reliable assessment of overall health status [26]. In this study, SAH was elicited through a single question: "In general, would you say your health is. . .." There were five response alternatives: *poor*, *fair*, *good*, *very good*, and *excellent*. According to respondents' self-perceived report, responses in these categories were 4.25%, 21.55%, 39.59%, 24.93%, and 9.68%, respectively. Since the frequency of responses in extreme categories, i.e., poor and excellent, were low, we merged the responses 'poor' and 'fair' as 'fair', and 'very good' and 'excellent' as 'excellent' during empirical analysis. SAH was then reclassified into three categories as: (a) fair = 1, if the respondent reports that her health is poor or fair; (b) good = 2, if the respondent reports that her health is good; and (c) excellent = 3, if the respondent reports that her health is very good or excellent. After recategorization, 25.81%, 39.59%, and 34.60% of the respondents fall into the fair, good, and excellent SAH categories, respectively. A broad range of SAH status determinants was included as explanatory variables in this study, including employment, possession of children, race, age, gender, physical activity, SNAP participation, educational level, store preference for buying food, and some interaction variables (i.e., *Race×Age*, *Race×Employment*, and *Employment×Children*). The description of the variables is given in Table 1.

**Table 1. Variable description and summary statistics.**

| Variables | Description | Unit | Frequency | Percentage (%) |
|---|---|---|---|---|
| *Response variable* | | | | |
| SAH | Self-assessed health status | Fair = 1, | 236 | 25.81 |
| | | Good = 2, | 176 | 39.59 |
| | | Excellent = 3 | 270 | 34.60 |
| *Explanatory variables* | | | | |
| Employment | Whether the respondent is employed (full or part-time) or not | Employed = 1, | 288 | 42.23 |
| | | Otherwise = 0 | 394 | 57.77 |
| Children | Whether the respondent has any children | Have children = 1, | 300 | 43.99 |
| | | No child = 0 | 382 | 56.01 |
| Race | Race of the respondent | Black or African American = 1, | 548 | 80.35 |
| | | Otherwise = 0 | 134 | 19.65 |
| Gender | Gender of the respondent | Female = 1, | 421 | 61.73 |
| | | Male = 0 | 261 | 38.27 |
| Exercise | Exercise frequency of at least 30 minutes per day | Yes = 1, | 485 | 71.11 |
| | | No = 0 | 197 | 28.89 |
| SNAP | Whether the respondent participated in the Supplemental Nutrition Assistance Program (SNAP) | Yes = 1, | 244 | 35.78 |
| | | No = 0 | 438 | 64.22 |
| Education | Highest educational degree | Less than high school = 1; | 99 | 14.52 |
| | | High school = 2; | 323 | 47.36 |
| | | Some college degree or vocational training = 3; and | 139 | 20.38 |
| | | Associate degree and above = 4 | 121 | 17.74 |
| Store choice | Most frequently visited store for grocery shopping | Other store = 1; | 66 | 9.68 |
| | | Jong's Super = 2; | 174 | 25.51 |
| | | Greensburg Market = 3; | 116 | 17.01 |
| | | Mac's Fresh Market = 4; and | 176 | 25.81 |
| | | Walmart or Winn-Dixie = 5 | 150 | 21.99 |
| | | | Mean | SD |
| Age | Age of respondent | Years | 50.83 | 16.99 |

## Empirical analysis

To answer our first research question that relates to assessing the factors influencing SAH, we used the ordinal logit regression model. Similarly, we used principal component analysis to answer our second research question about the construction of the obesity vulnerability index. A detailed explanation of our approach to empirical analysis is presented below.

## Ordered logit model

The dependent variable, SAH, is measured by three ranked or ordered categories. Therefore, we applied the ordered logit regression model to evaluate what factors affect the overall SAH status of an individual. Following Long and Freese [27], the specification of the ordered logit model can be expressed as

$$y_i^* = x_i'\beta + \varepsilon_i \tag{1}$$

where $y_i^*$ is a continuous latent variable representing SAH status for individual *i*. *x* is a vector

of explanatory variables, $\beta$ is a vector of parameters to be estimated, and $\varepsilon_i$ denotes independent and identically distributed (*iid*) error term.

Here, $y_i$ is the observed discrete, ordinal rating on a three-point scale for SAH status, i.e., $y_i$ = 1,2, *or* 3 for poor, fair, and excellent categories, respectively. $y_i$ is thus represented as

$$y_i = \begin{cases} 1 & \text{if } y_i^* \leq u_1 \\ 2 & \text{if } u_1 < y_i^* \leq u_2 \\ 3 & \text{if } y_i^* > u_2 \end{cases} \tag{2}$$

where $u_1$ and $u_2$ are known as threshold parameters (cut points) that can be estimated along with $\beta$ and $u_1 < u_2$. The model estimation is performed by the maximum likelihood method. The maximum likelihood estimator can be represented as [28]:

$$logL = \sum_{i=1}^{N} \sum_{j=1}^{J} y_{ij} log\left[ F(u_j - x_i'\beta) - F(u_{j-1} - x_i'\beta) \right], \tag{3}$$

where, $y_{ij} = \begin{cases} 1 \text{ if } y_i = j \\ 0 \quad else \end{cases}$ and Eq (4) is maximized with respect to $(\beta, u_1, \ldots, u_{j-1})$.

## Constructing obesity vulnerability index

The index-based method to assess vulnerability is quite popular in social and environmental research and can readily be applied in the public health sector too. OVI provides a single representative value for obesity exposure and preventive efforts of households or individuals that could help policymakers calculate community risk from obesity and formulate plans accordingly. We used indicators such as SAH, level of physical activity, food purchasing behavior, and demographic information to construct an index that singly represents an aggregate picture of the obesity problem in the study area. The details regarding indicator variables are provided in Table 2.

**Table 2. Description of indicator variables used in assessing vulnerability.**

| Category | Indicator variables | N | Mean | SD | Factor share |
|---|---|---|---|---|---|
| Sensitivity (Conducive factors) | Frequent shopping venue (Convenience or dollar store = 1, 0 otherwise) | 633 | 0.05 | 0.21 | 0.0541 |
| | Number of dependents | 630 | 2.80 | 1.72 | 0.6630 |
| | Number of vehicles | 626 | 1.52 | 1.08 | 0.2737 |
| | Pantry use (Yes = 1, 0 otherwise) | 621 | 0.25 | 0.44 | 0.0937 |
| | Age (years) | 622 | 50.69 | 17.11 | -0.6399 |
| | Gender (Female = 1, 0 otherwise) | 623 | 0.60 | 0.49 | 0.0602 |
| | Hispanic (Yes = 1, 0 otherwise) | 616 | 0.01 | 0.11 | 0.0880 |
| | Race (Black or African American = 1, 0 otherwise) | 613 | 0.79 | 0.41 | 0.2198 |
| | Food insecure (Yes = 1, 0 otherwise) | 607 | 0.30 | 0.46 | -0.0680 |
| Preventive efforts (Unconducive factors) | Bike access (Yes = 1, 0 otherwise) | 623 | 0.22 | 0.41 | 0.2882 |
| | Weekly exercise frequency (days) | 607 | 2.95 | 2.50 | 0.3423 |
| | Self-reported health (Good and above = 1, 0 otherwise) | 624 | 0.73 | 0.44 | 0.4978 |
| | Education level (Bachelor's degree or above = 1, 0 otherwise) | 604 | 0.15 | 0.35 | -0.0940 |
| | Employed (Yes = 1, 0 otherwise) | 605 | 0.42 | 0.49 | 0.4163 |
| | Sidewalk access (Yes = 1, 0 otherwise) | 611 | 0.24 | 0.43 | 0.4118 |
| | Safe traffic for walking and biking (Yes = 1, 0 otherwise) | 601 | 0.56 | 0.50 | 0.4475 |

Note: Factor share is the first component value that denotes the variable weight obtained from the Principal Component Analysis (PCA).

To make variables with different units comparable to each other, we standardized all the variables as

$$z = \frac{x_{ik} - \bar{x}_k}{s_{x_k}}$$

(4)

where $z$ denotes the standardized score, $x_k$ denotes $k^{\text{th}}$ indicator variable, $\bar{x}$ is the mean value, $s$ is the standard deviation, and $i$ indexes observations. Using standardized values, we run principal component analysis (PCA) to calculate the weight of each indicator used to construct the OVI. PCA is a popular non-parametric tool that allows the use of a wide range of indicators while also providing credible unequal weighting to the indicators [29, 30].

Before running the PCA, the Kaiser-Meyer-Olkin (KMO) test was conducted to examine the sampling adequacy of data. The KMO measure was 0.593, indicating that the data are satisfactory for PCA analysis. The index value was obtained by multiplying the component 1 score from PCA (Table 2) with the standardized value of each variable and summing them. The weights obtained from the PCA vary between -1 to +1 and the magnitude of the weights is an indication of the relative contribution of indicators to the OVI. We used the integrated vulnerability assessment framework of IPCC [31] with the REFLA survey data for calculating the OVI. The vulnerability of a respondent to obesity is defined as the net value obtained by subtracting the level of preventive efforts from the overall sensitivity level:

$$\text{Vulnerability} = \text{Sensitivity} - \text{Preventive efforts}$$

(5)

Based on this definition, the OVI can be expressed as

$$OVI = \sum_{k=1}^{9} x_{ik} w_{x_k} - \sum_{p=1}^{7} y_{ip} w_{y_p}, \; x \, \varepsilon \, \text{X}, \; y \, \varepsilon \, \text{Y}, \; \text{and} \; i = 1, \ldots, \text{N}.$$

(6)

where OVI denotes the obesity vulnerability index, and $x$ and $y$ refer to the sensitivity and preventive efforts indicators, respectively. $w$ stands for the weights obtained as first component loadings from principal component analysis for the $k^{\text{th}}$ or $p^{\text{th}}$ indicator, and $i$ indexes respondents. Assigning weights based on the first component values is widely practiced and reliably assigns weights to construct an empirically valid index [8, 32]. A higher value of OVI indicates a higher vulnerability to obesity and vice versa; however, this is not an absolute but rather a subjective measure to facilitate comparative ranking among sampled respondents. The resulting OVI provides insights for strategic planning and weighing alternatives for coping with obesity. All statistical analyses were conducted using Stata 17.

## Results

The summary statistics of the variables under consideration are presented in Table 1. Around 40% of the respondents reported SAH status as good (39.59%). Respondents' average age was 50.83 years (SD = 16.99). Most respondents were female (62%), Black (80%), unemployed (58%), had only a high school degree or less (61%), and did not have children (56%). Similarly, 71% of respondents reported having at least 30 minutes of daily physical activities. In addition, about 35% of respondents participated in the SNAP. Majority of the respondents indicated that they buy most of their food from local stores such as Jong's Super (26%), Greensburg Market (17%), and Mac's Fresh Market (26%). Only around 22% of respondents buy their food from wholesale chains such as Walmart or Winn-Dixie indicating that respondents mostly choose cheaper areas to buy food.

**Table 3. Estimated coefficients and marginal effects from the ordered logit model.**

| Variables | Odds ratio | Marginal effect (*dy/dx*) | | |
|---|---|---|---|---|
| | | *Dependent variable = Self-assessed health (SAH)* | | |
| | | Fair | Good | Excellent |
| *Employment* | 1.833 (0.700) | -0.04 (0.028) | -0.01 (0.008) | 0.06 (0.036) |
| *Children* | 1.771** (0.408) | -0.07** (0.031) | -0.02* (0.01) | 0.09** (0.04) |
| *Race (African American = 1)* | 6.998** (5.441) | -0.03 (0.037) | -0.005 (0.005) | 0.035 (0.042) |
| *Age* | 0.996 (0.011) | 0.005*** (0.001) | 0.001*** (0.0005) | -0.006*** (0.001) |
| *Gender (Female = 1)* | 0.693** (0.108) | 0.065** (0.027) | 0.015* (0.008) | -0.08** (0.034) |
| *Exercise frequency* | 1.596*** (0.268) | -0.08*** (0.029) | -0.02** (0.009) | 0.10*** (0.036) |
| *SNAP* | 0.819 (0.144) | 0.035 (0.031) | 0.008 (0.008) | -0.043 (0.038) |
| *Education* | | | | |
| High school | 1.282 (0.289) | -0.041 (0.047) | 0.002 (0.005) | 0.048 (0.042) |
| Some college or training | 2.107*** (0.548) | -0.13*** (0.048) | -0.03 (0.017) | 0.16*** (0.054) |
| Associate degree or above | 2.056*** (0.568) | -0.13*** (0.05) | -0.02 (0.018) | 0.15*** (0.058) |
| *Store choice* | | | | |
| Jong's Super | 0.527** (0.155) | 0.10** (0.042) | 0.05 (0.031) | -0.15** (0.069) |
| Greensburg Market | 0.388*** (0.120) | 0.161*** (0.05) | 0.046 (0.031) | -0.207*** (0.070) |
| Mac's Fresh Market | 0.641 (0.187) | 0.065 (0.04) | 0.04 (0.031) | -0.105 (0.07) |
| Walmart-Winn Dixie | 0.657 (0.195) | 0.06 (0.041) | 0.039 (0.032) | -0.10 (0.071) |
| *Race×Age* | 0.967*** (0.012) | 0.006*** (0.001) | 0.002*** (0.001) | -0.008*** (0.001) |
| *Race×Employment* | 0.806 (0.327) | -0.007 (0.050) | -0.003 (0.018) | 0.01 (0.068) |
| *Employment×Children* | 0.665 (0.205) | -0.026 (0.039) | -0.011 (0.017) | 0.037 (0.055) |
| Log-likelihood | -671.14 | – | – | – |
| LR chi$^2$(17) | 135.77*** | – | – | – |
| Pseudo R$^2$ | 0.092 | | | |
| *p*-value | 0.000 | | | |
| Number of observations | 682 | 682 | 682 | 682 |

Notes: Standard errors in parentheses.

***<0.01

**<0.05, and

*<0.10.

## Factors influencing SAH

One of the assumptions of the ordered logit model is the proportional odds or parallel regression assumption, which assumes that the relationship between each pair of outcome groups is the same. We test this assumption using the Brant test and the results from this test indicate that the parallel regression assumption holds ($\chi^2$ = value = 4.93; *p*-value >0.295). Similarly, the likelihood ratio test statistic is 135.77 ($p$ <0.001), indicating that the model fits well with the data. The odds ratio values of the variables obtained from the ordered logit model are presented in Table 3.

As shown in Table 3, the odds ratio of being in a better SAH category is significantly higher if the respondent has children, belongs to the Black or African American race, has some college or above education, and has a high exercise frequency of at least 30 minutes per day. However, the odds ratio of being in a better SAH category is significantly lower if the respondent is female and goes to local grocery stores such as Jong's Super and Greensburg Market to buy food.

Although the odds ratio is informative about the influence of a variable on SAH, it does not give precise information about how changes in the explanatory variables affect SAH. Thus, the most natural way to interpret an ordered response model is to determine how a marginal change in one explanatory variable changes the distribution of the response variable [33]. The marginal effect indicates how the probabilities of being in a particular category of SAH change as we vary one variable and hold the remaining variable at their means. The marginal effects of the explanatory variables on SAH are also reported in Table 3.

The results indicate that variables such as *having children*, *age*, *gender*, *exercise frequency*, *education level*, *store choice*, and the interaction of *race* and *age* significantly affect SAH rating as good. However, *employment status*, *race*, *SNAP participation*, *Race×Employment*, and *Race×Children* have no significant effects on the probability of reporting SAH as fair or excellent.

Having children is associated with a 7% and 2% decline in the probability of respondents rating their SAH in the fair and good categories, respectively, compared to those having no children. While the likelihood of a SAH rating as excellent increased by 9% for respondents with children. Each additional year of age increases the probability of reporting SAH as fair by 0.5% and good by 0.1%, but the chance of reporting excellent decreases by 0.6%. Similarly, probabilities of SAH being fair and good were respectively 6.5% and 1.5% higher for females compared to males while the probability of reporting excellent SAH reduced by 8% for females.

The marginal effect of physical activity on SAH shows that at least 30 minutes of exercise per day was associated with a decrease of 8% and 2% in the probabilities of SAH status as fair and good, respectively; however, the likelihood of an excellent SAH status increased by 10%. Moreover, the marginal effect of education level indicates that the probabilities of SAH rating as excellent increased by 16% and 15% for those who have some college training and an associate degree or above, respectively, compared to those who have a high school or less education level.

Regarding the food purchasing behavior, buying most of the food from local stores like Jong's Super store was associated with an increase of 10% in the probabilities of SAH rating as fair, compared to those who buy most of their food from other stores; however, the likelihood of a SAH rating as excellent was reduced by 15%. Similarly, buying most of the food from another local market, Greensburg Market, was associated with a 16.1% increase in the probabilities of SAH rating as fair; however, the chance of reporting excellent SAH decreased by 20.7%.

The estimated marginal effect for the race-by-age interaction term indicated that probabilities of SAH ratings of fair and good were 0.6% and 0.2% higher, respectively, for each increment in years of age among Black or African Americans than Whites and others, while the chance of reporting excellent decreased by 0.8%.

## Obesity vulnerability index estimates

The respondents in rural Louisiana have a mean OVI of 0.006, but ranges from -4.036 to 4.565, indicating a higher level of vulnerability to obesity. The distribution of OVI for respondents is presented in Fig 2. The figure shows a wide discrepancy in the vulnerability level of individuals regarding obesity.

To get more insights about the vulnerability to obesity of respondents, we further developed ordered quintiles of vulnerability categories: very low, low, medium, high, and very high. The frequency statistics of the vulnerability category are presented in Table 4. Around 20% of the respondents fall into the most-vulnerable segment, and ~65% of respondents are vulnerable to obesity either being in the medium, high, or very high category.

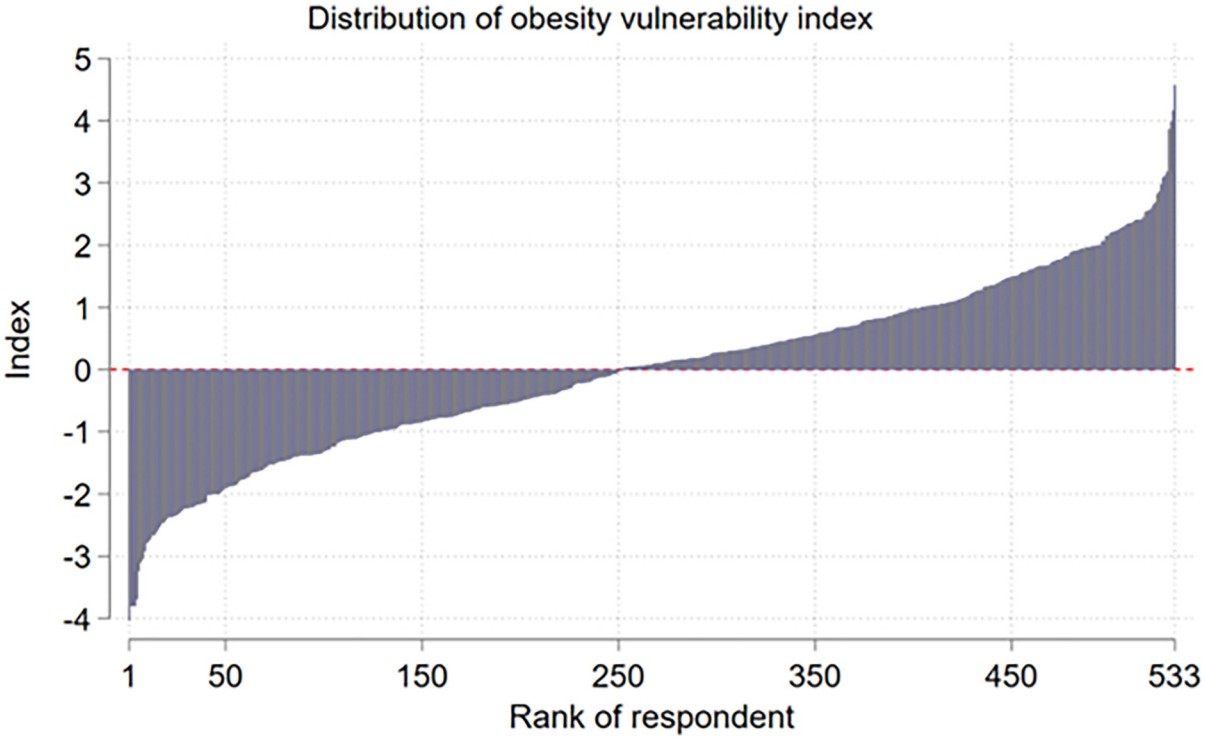

**Fig 2. Distribution of OVI among respondents in the study area.** Note: The rank of the respondent is sorted in an ascending order based on index values.

The breakdown of OVI by variable categories is described in Table 5. Among the three counties in this study, Saint Helena (OVI = 0.291) is the most vulnerable to obesity, followed by East Carroll (OVI = 0.106) and Tensas (OVI = -0.414). Similarly, the 18–40 age category individual seems to be much more vulnerable to obesity (OVI = 0.781) compared to other age groups. Similarly, women (OVI = 0.150) are more vulnerable to obesity than men (OVI = -0.219). Furthermore, Black or African American respondents (OVI = 0.221) are more vulnerable to obesity than others. Similarly, having one or two bikes at home is associated with lower OVI; however, OVI is higher with the number of bikes >3.

## Discussion

The findings from the study imply that grocery store choice and exercise frequency have significant associations with the overall SAH status, besides demographic factors such as gender,

**Table 4. Frequency distribution of vulnerability categories.**

| Category | Frequency | Percentage (%) |
|---|---|---|
| Very high | 9 | 1.69 |
| High | 102 | 19.14 |
| Medium | 238 | 44.65 |
| Low | 158 | 29.64 |
| Very low | 26 | 4.88 |
| Total | 533 | 100 |

**Table 5. Obesity vulnerability index summary by different categories of variables.**

| Variable | Mean | Variable | Mean |
|---|---|---|---|
| *County* | | *Exercise (days per week)* | |
| East Carroll | 0.106 | 0 | 0.594 |
| Saint Helena | 0.291 | 1 | 0.074 |
| Tensas | -0.414 | 2 | 0.071 |
| *Age category* | | 3 | -0.138 |
| 18–40 years | 0.781 | 4 | -0.084 |
| 41–60 years | -0.062 | 5 | -0.240 |
| 60+ years | 0.626 | 6 | -0.792 |
| *Gender* | | 7 | -0.663 |
| Female | 0.150 | *Number of bikes* | |
| Male | -0.219 | 0 | 0.060 |
| *Race* | | 1 | -0.339 |
| Black or African American | 0.221 | 2 | -0.182 |
| Others | -0.83 | 3 or above | 0.672 |
| *Self-assessed health* | | *Number of vehicles* | |
| Poor | 0.892 | 0 | -0.147 |
| Fair | 0.713 | 1 | -0.048 |
| Good | -0.369 | 2 | -0.046 |
| Very good | 0.036 | | |
| Excellent | -0.575 | | |

education, and age-race interaction. This is not unusual because local grocery stores provide few food choices compared to big box grocery stores. The negative influence of local grocery store choice implies that lower-income families have limited access to supermarkets and other healthy food retail outlets that provide varieties of affordable and nutritious foods. These limitations with local stores may be associated with poorer dietary choices and consequently reporting poor SAH. Similarly, rural infrastructure does not well support walking, running, or biking activities, thus exercise frequency is likely to be hampered that have implications for the SAH status of rural residents.

The significant value of race by age interaction term suggests that racial disparity in SAH is affected by the age of the respondent. In other words, with every additional year of age, Black respondents are less likely to value their SAH to be in the better categories. An unexpected result is a significantly positive association between race and SAH. A possible explanation is that more than 80% of respondents are black or African American, and almost 74% of these respondents reported their health status as good or excellent, which is largely reflected in the SAH. The positive association of having children with better SAH is in line with the findings by Fritzell and Gähler [34].

The marginal effects for age had a significant and positive impact on the 'fair' and 'good' levels of health status but a negative sign on the 'excellent' level. In general, older people may have more disabilities and compromised health conditions that lead to poor SAH ratings compared with younger adults. The results are consistent with previous studies, including Andersen et al. [35], McFadden et al. [36], and Jurewicz and Kaleta [37].

Generally, regular physical activity is one of the most important things people can do to improve their health. The more often an individual is physically active, the better their health status. The results are in line with previously published studies that suggested a positive relationship between physical activity and SAH [38, 39]. The significantly positive marginal effect of higher educational attainment on the excellent SAH category may be because educational

attainment is associated with better health [12, 40]. Regarding the marginal effects of race, our results are consistent with the findings by Lee et al. [41] and Krok-Schoen et al. [42]. The authors found that older African Americans were more likely to rate SAH as poor. The Centers for Disease Control and Prevention [43] also reported that Black adults are more likely to report their general health status as fair or poor compared to White adults.

The results relating to OVI show that more than 65% of the respondents fall into vulnerable segments, either in the moderate, high, or very high category. Such a high level of vulnerability is an indication of the severity of obesity-related risks in the study area, thus prompting policy actions to cope with obesity in the study area. Furthermore, the OVI estimates suggest that the respondents with SAH as good, very good, and excellent are less vulnerable to obesity than those reporting fair or poor. This implies that respondents' self-assessment of their health status is correct. The vulnerability to obesity decreased with the increasing frequency of exercise. The rate of decrease of OVI with 30-minute exercise is steeper up to 4 days per week, after which it starts to increase. This U-shaped relationship between exercise and OVI is interesting but unclear, thus needs further investigation. Having more vehicles in the home is also associated with higher OVI. One unusual observation from our study is that OVI is increasing when the number of bikes in a household is >3. This needs additional investigation and might imply that using bikes is more important than merely having them. Having sidewalks on roads and the perception that traffic is safe for walking or biking around the residential area is associated with lower OVI. This has implications for Louisiana, where sidewalks are less common, and the crime rate is higher than in other states in the US. The results further imply that there is wide heterogeneity in the distribution of OVI by different categories of variables. Thus, targeting programs could be more useful in combating the obesity-related problem.

Policymakers and health practitioners need credible evidence before making decisions to address obesity in rural communities. This study uses representative data to determine risk and protective factors for obesity that could inform evidence-based interventions for decision-makers. First, results indicating that low-income households are more at risk for obesity reinforce the need for evidence-based nutrition and physical activity interventions targeted at low-income rural residents. Second, results show a lower risk of obesity associated with the presence of sidewalks and perceptions of safe walking and biking support policies and funding that improve pedestrian infrastructure and rural road design to encourage active transportation in rural communities. Lastly, the associations between grocery store choice and obesity vulnerability support policy and interventions that encourage improved access to healthy foods, both in terms of availability and affordability, in the rural food retail sector.

Although our study sheds light on how different factors influence SAH and the underlying vulnerability to obesity among rural residents, there are a few caveats with our analyses. First, only three Louisiana counties were included, which might limit the generalization of the study; however, these counties are still the priority of the Centers for Disease Control (CDC) programs for preventing obesity. Second, we consider socioeconomic and health-related factors in constructing OVI; however, further research using more comprehensive data about actual diet choices could strengthen our findings. Third, survey respondents may not always have been the primary food purchaser of the household which could have impacted responses to questions about food shopping behaviors and experiences. Lastly, the inferences drawn using cross-sectional data could be bolstered by using household-level longitudinal data.

## Conclusions

Obesity is a rapidly emerging challenge for public health practitioners in the rural US. Thus, the information about the correlates of SAH status and the underlying vulnerability of the

communities to obesity is critical to addressing burgeoning obesity-related problems. In this study, we used representative survey data from rural Louisiana to examine the socio-economic factors influencing SAH and calculated the OVI for low-income communities. The results suggest that interventions to increase grocery store choice and exercise infrastructure could promote SAH, thus lowering vulnerability to obesity. Since, majority (~65%) of the residents fall in the obesity-vulnerable segment, rolling out targeted prevention and control measures is very necessary to minimize the rural-urban gap in obesity prevalence.

Our findings further show that there is a wide heterogeneity in the vulnerability level of the rural residents, thus necessitating diversity in intervention groups instead of a blanket approach while implementing obesity prevention and control measures. The findings of this research provided insights into a policy discussion about designing an effective and efficient suite of interventions in low-income communities to fight obesity. As currently structured, only a few low-income communities are included in this study with an exclusive focus on northern Louisiana. Future studies could include all low-income communities to evaluate how individuals' SAH level changes along with their socio-economic characteristics at a regional or state level. OVI estimates could be further strengthened by a more extensive study incorporating a broader range of variables at both household and community levels. All these topics are left for future research.

## Supporting information

**S1 Data.**
(DTA)

**S1 File.**
(DOCX)

## Acknowledgments

The authors are thankful to the LSU Agricultural Center, Marquetta Anderson, Joy Sims, Cecilia Stevens, Makenzie Miller, Matt Greene, Ruthie Losavio, Bailey Houghtaling, Nila Pradhananga, Judith Rhodes, Rene Lavinghouse, Charlymane McCray, Toni Melton, Joetta Shields-Pitts, Rebekah Rodriguez, Jenna Wehner, community data collectors, and RELFA survey respondents.

## Author Contributions

**Conceptualization:** Katherine Seals, Denise Holston.

**Data curation:** Katherine Seals, Denise Holston.

**Formal analysis:** Santosh Pathak, Hua Wang, Naveen C. Adusumilli.

**Funding acquisition:** Katherine Seals, Denise Holston.

**Methodology:** Santosh Pathak, Hua Wang.

**Project administration:** Katherine Seals, Denise Holston.

**Resources:** Naveen C. Adusumilli, Denise Holston.

**Software:** Santosh Pathak, Hua Wang.

**Supervision:** Naveen C. Adusumilli, Denise Holston.

**Validation:** Hua Wang, Naveen C. Adusumilli, Denise Holston.

**Writing – original draft:** Santosh Pathak, Hua Wang, Naveen C. Adusumilli.

**Writing – review & editing:** Santosh Pathak, Hua Wang, Katherine Seals, Naveen C. Adusumilli, Denise Holston.

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
