## [Decision Letter · Decision Letter 0]

4 Nov 2022

PONE-D-22-25232Self-assessed health status and obesity vulnerability index of low-income families in northern LouisianaPLOS ONE

Dear Dr. Pathak,

Thank you for submitting your manuscript to PLOS ONE. After careful consideration, we feel that it has merit but does not fully meet PLOS ONE’s publication criteria as it currently stands. Therefore, we invite you to submit a revised version of the manuscript that addresses the points raised during the review process.

We look forward to receiving your revised manuscript.

Kind regards,

Larissa Loures Mendes, Ph.D.

Academic Editor

PLOS ONE

Journal Requirements:

2. In the ethics statement in the Methods, you have specified that verbal consent was obtained. Please provide additional details regarding how this consent was documented and witnessed, and state whether this was approved by the IRB.

3. Please include your ethics statement in the Methods section of your manuscript. In the Methods section of your revised manuscript, please include the full name of the institutional review board or ethics committee that approved the protocol, the approval or permit number that was issued, and the date that approval was granted.

Additional Editor Comments:

Dear Authors,

Unfortunately, in the current format the article cannot be considered for publication. The introduction of the study is confusing, in the methods the statistical analyses need to be detailed, and the presentation of the results and discussion is very confusing. Also, the conclusion needs to be revised. The reviewers made several suggestions and I believe that after an extensive revision the study can be considered for publication again.

Sincerely,

Larissa Loures Mendes

Reviewers' comments:

Reviewer's Responses to Questions

**Comments to the Author**

1. Is the manuscript technically sound, and do the data support the conclusions?

Reviewer #1: Partly

Reviewer #2: No

2. Has the statistical analysis been performed appropriately and rigorously? 

Reviewer #1: No

Reviewer #2: No

3. Have the authors made all data underlying the findings in their manuscript fully available?

Reviewer #1: Yes

Reviewer #2: Yes

4. Is the manuscript presented in an intelligible fashion and written in standard English?

Reviewer #1: No

Reviewer #2: Yes

5. Review Comments to the Author

Reviewer #1: I congratulate the authors for the work developed. During the reading of the paper I had some doubts that are listed below.

- I suggest that the authors leave the sections of the article more delimited, in some moments there are questions about methodology presented in the results and vice-versa.

- Introduction: It is not entirely clear what is the gap in the literature that the study will answer. Once the data presented in the introduction demonstrates that the factors that influence obesity are already well described in the literature. I suggest structuring the introduction more directly and making the justification for the work clearer.

- What criteria were used to select the three rural communities included in the study?

- What were the inclusion and exclusion criteria for the participants? It would be important to make clear what criteria were adopted in conducting the study.

- What software was used to perform the statistical analyses?

- In the results it appears that the chi-square test was performed, but this type of information is not available in the methodology. It would be important to make clear in the methodology all the statistical analyses that were performed.

- The discussion needs to be reformulated, because the authors focused too much on presenting that the results agree with previously developed studies. It would be important to provide more robust explanations for the results found.

Reviewer #2: I appreciate the opportunity to evaluate the article in question.

This is an article that sought to investigate the influence of sociodemographic factors on self-rated health and determine vulnerability to obesity in rural communities.

This is an important theme for public health and a special one for working with a population group outside the urban context.

I point out some necessary changes in the article:

SUMMARY - the objective needs to be direct. I suggest deleting the first sentence of the objective.

The methods need to bring detail beyond PCA analysis.

The conclusions presented are final considerations. I suggest a rewrite.

INTRODUCTION -

It is too long. I think it deserves a restructuring. The research problem is reduced in the introduction and should be more detailed.

Until line 61, the introduction could be reduced to a single paragraph.

Os objetivos do resumo e introdução estão diferente, sugiro a padronização.

METHODS

I believe that the initial part of the methods is missing. It is not presented with the methodological design of the study - I believe it is a cross-sectional study. This needs to be informed. I suggest writing it according to STrengthening the Reporting of OBservational studies in Epidemiology - STROBE.

Regarding PCA, I suggest explaining the process of analysis better. Was it measured eigenvalues > 1.0, defined according to the scree plot for the extraction of the components?

This is not clear

Why was the coefficient and not the OR presented?

RESULTS AND DISCUSSION I think there was a mistake in the writing. I suggest separating results from discussion.

I suggest presenting the PCA analysis in the results separately.

The tables need to be redone - they present the code of the variables, but I believe that this is not necessary for this article.

Presenting the discussion this way ended up compromising the depth of the discussion, becoming superficial and not evolving the results found.

The study does not present the limitations, I suggest its inclusion.

CONCLUSION

The way it is presented it is not a conclusion of the study but final considerations. I suggest rewriting the conclusions.

6. PLOS authors have the option to publish the peer review history of their article (what does this mean?). If published, this will include your full peer review and any attached files.

Reviewer #1: No

Reviewer #2: No

---

## [Author Response · Author response to Decision Letter 0]

19 Dec 2022

Response to the reviewers file is attached separately.

---

## [Decision Letter · Decision Letter 1]

1 Jun 2023

Self-assessed health status and obesity vulnerability in rural Louisiana: A cross-sectional analysis

PONE-D-22-25232R1

Dear,

We’re pleased to inform you that your manuscript has been judged scientifically suitable for publication and will be formally accepted for publication once it meets all outstanding technical requirements.

Kind regards,

Fernanda Penido Matozinhos, Ph.D

Academic Editor

PLOS ONE

Additional Editor Comments (optional):

Dear authors, the manuscript explores a very important topic and it has technical rigor. Thank you for submitting your manuscript to PLOS ONE and making substantial changes in order to improve the manuscript. I congratulate the authors for the work developed. The objective is relevant and the results are of interest for a wide range of potential readers. I recommend its publication.

Reviewers' comments:

Reviewer's Responses to Questions

**Comments to the Author**

1. If the authors have adequately addressed your comments raised in a previous round of review and you feel that this manuscript is now acceptable for publication, you may indicate that here to bypass the “Comments to the Author” section, enter your conflict of interest statement in the “Confidential to Editor” section, and submit your "Accept" recommendation.

Reviewer #1: All comments have been addressed

2. Is the manuscript technically sound, and do the data support the conclusions?

Reviewer #1: Yes

3. Has the statistical analysis been performed appropriately and rigorously? 

Reviewer #1: Yes

4. Have the authors made all data underlying the findings in their manuscript fully available?

Reviewer #1: Yes

5. Is the manuscript presented in an intelligible fashion and written in standard English?

Reviewer #1: Yes

6. Review Comments to the Author

Reviewer #1: I would like to congratulate the authors for their care in responding to comments. The current form of the manuscript has made the purpose of the study clearer and the discussion more robust.

7. PLOS authors have the option to publish the peer review history of their article (what does this mean?). If published, this will include your full peer review and any attached files.

Reviewer #1: No

---

## [Editor Report · Acceptance letter]

9 Jun 2023

PONE-D-22-25232R1 

Self-assessed health status and obesity vulnerability in rural Louisiana: A cross-sectional analysis 

Dear Dr. Pathak:

I'm pleased to inform you that your manuscript has been deemed suitable for publication in PLOS ONE. Congratulations! Your manuscript is now with our production department. 

Kind regards, 

on behalf of

Dr. Fernanda Penido Matozinhos 

Academic Editor

PLOS ONE